# Micronutrient Status of Electronic Waste Recyclers at Agbogbloshie, Ghana

**DOI:** 10.3390/ijerph17249575

**Published:** 2020-12-21

**Authors:** Sylvia A. Takyi, Niladri Basu, John Arko-Mensah, Duah Dwomoh, Afua Asabea Amoabeng Nti, Lawrencia Kwarteng, Augustine A. Acquah, Thomas G. Robins, Julius N. Fobil

**Affiliations:** 1Department of Biological, Environmental & Occupational Health Sciences, School of Public Health, University of Ghana, Accra +233, Ghana; jarko-mensah@ug.edu.gh (J.A.-M.); effya76@gmail.com (A.A.A.N.); lkwarteng@st.ug.edu.gh (L.K.); nana.austine@gmail.com (A.A.A.); jfobil@ug.edu.gh (J.N.F.); 2Faculty of Agricultural and Environmental Sciences, McGill University, Montreal, QC H9X 3V9, Canada; niladri.basu@mcgill.ca; 3Department of Statistics, School of Public Health, University of Ghana, Accra +233, Ghana; duahdwomoh@gmail.com; 4Environmental Health Sciences, School of Public Health, University of Michigan, Ann Arbor, MI 48109, USA; trobins@umich.edu

**Keywords:** e-waste recyclers, micronutrients, pollution, nutrition, exposure assessment, biomarkers, informal sector, cross-sectional study

## Abstract

Growing evidence suggests that micronutrient status is adversely impacted by toxic metals (e.g., cadmium, lead, and arsenic) exposures; however, the micronutrient status of e-waste recyclers who are amongst the highest metal-exposed groups is not known. This study, therefore, assessed the micronutrient status of e-waste recyclers using dietary information (2-day 24-h recall survey) and biomarker data (whole blood and urine) among 151 participants (100 e-waste recyclers at Agbogbloshie and 51 controls at Madina Zongo from the Accra region, Ghana) in March 2017. Biomarker levels of iron (Fe), calcium (Ca), magnesium (Mg), selenium (Se), zinc (Zn) and copper (Cu) were analyzed by the ICP-MS. Linear regression models were used to assess associations ofwork-related factors and sociodemographic characteristics with micronutrient intake, blood, and urine micronutrient levels. The results showed that apart from Fe and Zn, e-waste recyclers at Agbogbloshie did not meet the day-to-day dietary requirements for Ca, Cu, Se, and Mg intake. Except for the low levels of Mg and Fe detected in blood of e-waste recyclers, all other micronutrients measured in both blood and urine of both groups fell within their reference range. Exposure to biomass burning was associated with lower blood levels of Fe, Mg, and Zn among the e-waste recyclers. Further, among e-waste recyclers, significant relationships were found between the number of years spent recycling e-waste and urinary Ca and Cu excretion. Given that, some dietary and blood levels of micronutrients were below their reference ranges, the implementation of evidence-based nutrition strategies remains necessary among e-waste recyclers to reduce their risk of becoming malnourished.

## 1. Introduction

Adequate consumption of foods rich in micronutrients such as calcium (Ca), magnesium (Mg), iron (Fe), copper (Cu), zinc (Zn), and selenium (Se) is essential for normal physiological functions [1]. Deficiencies in such micronutrients, perhaps owing to poor food intake, could precipitate adverse health conditions such as anemia, hypertension, diabetes, and other physiological impairments. Further, exposure to toxic metals such as cadmium (Cd), arsenic (As), and lead (Pb) may promote micronutrient deficiency, and in turn, this deficiency may aggravate the toxicity of metals. While some nutritional deficiencies may not be acutely life threatening, they may increase one’s susceptibility to chronic diseases. Given that both micronutrient deficiencies [1] and toxic chemical exposures [2] are major contributors to the global burden of disease, investigating their interactions particularly in vulnerable groups is of utmost importance.

The recycling of electronic waste (e-waste) is growing worldwide [3,4]. It is largely an informal sector activity situated in low- and middle-income countries, thus drawing from populations who likely experience micronutrient deficiencies [5]. The job is physically demanding; thus, e-waste recyclers need adequate nutrient intake. Many studies have established that these recyclers are exposed to many toxic chemicals [6,7,8,9]. The interactions between micronutrient and toxic chemicals including heavy metals are not fully known, albeit these are likely to impact on each other. Toxic metals and absorbed micronutrients may interact at several points in the body (i.e., during absorption, excretion, transport, or binding to target proteins). For instance, Pb may mimic Ca by entering voltage-activated L-type Ca channels in place of Ca [10,11], and Cd exhibits an inhibitory effect on Fe metabolism and absorption, which may cause decreases in hematocrit and hemoglobin levels [12,13]. In addition, several toxic metals may interact with micronutrients in blood, causing the displacement and subsequent excretion of micronutrients in urine, [14], feces [15], and sweat [16]. Micronutrient deficiencies may potentiate the toxicity of hazardous metals in the body [17,18]. Consistent with this observation, adequate nutrition also serves as an essential factor in preventing toxic metal poisoning [19], as poor nutrition status impairs the body’s defense system against exposure to toxic metals [20]. Several studies have investigated metals exposure among e-waste recyclers [6,7,8,9]. However, no attention has been given to investigating the micronutrient status of e-waste recyclers, despite the fact that they are continuously exposed to toxic metals and other environmental contaminants believed to negatively affect nutritional status [21,22,23,24].

In Ghana, the primary e-waste site is Agbogbloshie situated in central Accra [5] with several thousand recyclers dependent upon the industry [25]. Past studies have characterized relatively high exposures of recyclers at Agbogbloshie to a diverse range of chemicals [6,7], yet to our knowledge, no assessment of their micronutrient status has been done. We, therefore, addressed this critical knowledge gap by conducting a cross-sectional study at Agbogbloshie to (i) assess micronutrient (Ca, Mg, Se, Fe, Zn, and Cu) intake (based on dietary surveys), (ii) measure the levels of these micronutrients in the blood and urine, (iii) assess the relationship between dietary micronutrient intake and the micronutrient levels in blood and urine, and (iv) examine the associations of work-related factors and sociodemographic characteristics on dietary micronutrient intake, as well as the micronutrient levels in the blood and urine of the e-waste recyclers and controls.

## 2. Materials and Methods

### 2.1. Data Source and Study Design

Our study data were drawn from the GEOHealth-II longitudinal cohort study [26,27,28,29], which sought to address several outstanding questions concerning environmental and occupational health issues related to e-waste activities. Our cohorts (e-waste recyclers and controls) were selected based on their exposure status relative to the job tasks undertaken. The e-waste recyclers who live and work at the Agbogbloshie e-waste site are widely known to be exposed to a myriad of toxic chemicals [6,7,23,24], therefore served as our exposed group. In contrast, the control group (Madina-Zongo) live about 10 km away from the exposure site, include a population with similar demographics, and have never been involved in e-waste recovery. Using Diggle et al. [30] formula for longitudinal cohort analysis, the baseline (e-waste recyclers and controls) was calculated for the broader project. This resulted in a sample size of 92 e-waste recyclers and 40 controls. This sample size was increased to the final sample size of 100 e-waste recyclers (from the Agbogbloshie site) and 51 controls (from Madina, a community about 10 km south of Agbogbloshie), all in Accra in March 2017. The 100 e-waste recyclers included 32 burners, 49 dismantler, and 19 collectors/sorters whose primary job responsibility was to manage either the burning, dismantling, or the collecting/sorting of obsolete electronic equipment. For this micronutrient assessment study, we restricted our analyses to the baseline data obtained.

As detailed in our previous studies at Agbogbloshie [6,7,26,27], initially we organized a community durbar to inform, and also familiarize, eligible participants with the study’s objectives and procedures. A list of recyclers and controls who were willing to volunteer in the study was then generated after the durbar held at the various study sites in February 2017. For the e-waste recyclers, their list was categorized per their job task performed by the help of the Chairman and Vice-chairman of the e-waste recyclers at the initiation of recruitment. The inclusion criteria were adult males aged 18 years and above who had worked at the e-waste site for at least six months. Similarly, participants from the control site (Madina-Zongo, Accra, Ghana) were in the same age category and with similar demographic characteristics as the e-waste recyclers (e.g., culture and dietary habits). In addition, participants from the control site must have lived there for at least six months. We also considered e-waste recyclers who functioned under the management and control of the Chairman for recyclers to allow effective follow-up. Meanwhile, persons with mental or physical disabilities interfering with their ability to understand the informed consent or complete health status measures were not included as participants. Informed consent was obtained from all participants, and they were compensated with 50 Ghanaian cedis (approximately $10 USD, which is roughly an average day’s wage), a lunch pack and T-shirt. Institutional Review Boards at the University of Ghana, the University of Michigan and McGill University approved the study protocols. In addition, the local chief of Agbogbloshie and Madina-Zongo permitted our research team to enter the community to conduct this study.

### 2.2. Study Site

Agbogbloshie is an informal e-waste dumping area, scrap processing and food market zone located on the bank of the Korle-Lagoon, to the west of the Odaw River in the heart of Accra. The site is geographically hemmed within the areas of Abossey Okai road, the Odaw river and the Agbogbloshie drain. Nearly all the e-waste recyclers are Muslims from either the Dagomba or Konkomba ethnic groups, who migrated from the northern part of Ghana in search of improved employment opportunities. The Odaw river marks the border of the e-waste site and Old Fadama slum, which houses about 80,000 people [31,32].

This study also included a control group residing in Madina-Zongo who have similar socio-demographic characteristics with the e-waste recyclers with respect to religion and internal migration from northern parts of Ghana. Madina-Zongo is located within 10 km of the Agbogbloshie e-waste site and is expected to be unexposed to e-waste activities. These controls were engaged in several occupations, spanning from sewing to construction, security, teaching, and merchandising of general goods. Meanwhile, the possible similar exposures among the controls (compared to the recyclers) may likely emerge from uncontrolled biomass burning, vehicular emissions and dust from unpaved surfaces.

### 2.3. Field Data Collection Procedures

#### 2.3.1. Anthropometric Measurements

The participant’s height and weight were measured using a standardized protocol. Height was measured with a Seca stadiometer (Seca, Hamburg, Germany), and corrected to the nearest 0.1 cm, with participant standing upright on a flat surface without shoes, and the back of the heels, and the occiput against the stadiometer [33,34,35]. Further, participant’s body weight was measured to the nearest 0.1 kg using a portable Seca scale (Seca 770, Hamburg, Germany). The same model of a standard calibrated balance was used at both study sites. The body mass index (BMI) of the participants was calculated by dividing their weight in kilograms (kg) by height in meters squared (m^2^).

Based on previous exposure assessments and occupational health surveys conducted in Ghana [6,7,33], a semi-structured questionnaire was designed, piloted, and used to collect information on the sociodemographic characteristics and work history of the participants.

#### 2.3.2. Dietary Micronutrient Intake Assessment

Data collection took place over a period of two weeks in each study site. Daily nutritional intake of the participants was collected using a semi-structured 2-day 24-h recall guide. We conducted the 24-h recall twice to estimate the day-to-day variability that occurs per individual owing to the variety of foods consumed on different days. In an effort to maximize the consistency of the interview format across the study sites and also minimize between site methodological biases, trained dieticians were used as interviewers. The interview was conducted in local dialects of the participants’ native or preferred language—Dagbani, Hausa, Twi, or English—to ensure that the participants fully understood the questions asked with respect to the 2-day 24-h recall guide. The interview consisted of questions on foods and beverages (e.g., the amount, the time and the types of meals/foods) consumed on one weekday and one day of preceding weekend (Saturday or Sunday). In all cases, information was solicited within less than 24 h when that day ended. The dietary interview was done either face to face or through phone calls. We also used graduated food models [36,37] to quantify foods and beverages consumed by each participant.

#### 2.3.3. Blood and Urine Sample Collection

Urine and whole blood samples were collected in a clean and enclosed area near each of the study sites. A midstream urine sample (15 mL) was collected into a plastic container at the start of each participant’s visit to the clinic, typically between the hours of 9 a.m. and 4 p.m. Approximately 10 mL of blood was collected into a trace metal-free BD Vacutainer tube with K_2_EDTA, and placed on a blood tube roller (Micro-Teknik) for five minutes. The urine and blood samples were stored in a −80 °C freezer until shipment to McGill University (Montreal, Canada) where they were stored frozen (−80 °C) prior to analyses. We note that the laboratory analysis at McGill University was mainly performed by visiting African scholars as part of exchanges facilitated by the Geo-Health II project that aimed to increase research and technical capacity.

### 2.4. Data Analysis

#### 2.4.1. Dietary Micronutrient Intake Analysis

The dietary micronutrient intake data collected was converted into grams using Ghanaian food composition tables. We further undertook a comprehensive nutrient analysis using the ESHA F Pro^®^ software to estimate individual micronutrient intake. Data obtained from the ESHA F Pro after nutrient analysis consisted of Ca, Mg, Fe, Zn, Cu, and Se. These results were compared to the recommended daily allowance (RDA) for adult males [38].

#### 2.4.2. Laboratory Analysis of Micronutrients in the Blood and Urine

Micronutrients—namely, Ca, Mg, Fe, Se, Cu, and Zn—were analyzed in both whole blood and urine. The urine and blood samples were acid-digested as outlined in previous studies [6,39]. Elemental concentrations were determined using an inductively coupled plasma mass spectrometer (ICPMS; Varian 820MS). Several analytical quality control measures were used. All tubes and pipette tips used were acid-washed (cleaned, soaked for 24 h. in 10% hydrochloric acid and rinsed three times in Milli-Q water) prior to usage. Certified standard reference materials (INSPQ; QM-U-Q1109 [urine]; and then QM-B-Q1506 and QM-B-Q1314 [blood]) obtained from the Institut National de Santé Publique du Québec were used to measure accuracy and precision. Furthermore, each batch run consisted of procedural blanks and replicates. For each micronutrient analyzed, the theoretical detection limit (µg/L) was calculated as three times the standard deviation of the mean blank value (Appendix A).

### 2.5. Statistical Analysis

Dietary micronutrient intake (Ca, Mg, Fe, Zn, Cu, and Se), participant work status (e-waste recycler vs. control), job task (dismantlers, burners, and collectors/sorters), work characteristics (e.g., daily duration (hours) of e-waste work activity and number of years spent recycling e-waste), and other sociodemographic characteristics (e.g., educational status, daily income earned, and age) were our primary exposures of interest. Environmental risk factors such as cigarette smoking, exposure to biomass burning and alcohol intake were also notable exposures of interest. Depending on the model investigated, outcome variables were either micronutrient levels (Ca, Mg, Fe, Zn, Cu, and Se) in blood and urine or dietary micronutrient intake values.

Data analysis focused on the following a priori questions. First, to characterize dietary micronutrient intake, the fraction of e-waste recyclers and the controls who consumed adequate amounts of micronutrients was assessed by defining and dichotomizing each of the micronutrients based on the United States Department of Agriculture (USDA) guidelines [40] for adults. This definition outlines the threshold for micronutrient adequacy using data obtained from the micronutrient intake. The Wilcoxon rank-sum test was used to determine whether differences existed between the amounts of micronutrients consumed from diet by the e-waste recyclers and the controls. We further used the Kruskal–Wallis test to assess the differences in median dietary micronutrient intake levels among the e-waste recycler groups (burners, dismantlers, and collectors/sorters). Second, to compare the median differences in micronutrients measured in the whole blood and urine between the e-waste recyclers and controls, the Wilcoxon rank-sum test was used. We further used the Kruskal–Wallis test to examine the median differences in biomarker micronutrient levels measured between the e-waste recycler groups. Third, relationships between dietary micronutrient intake and micronutrient levels in blood and urine were gauged using Pearson’s correlation. Fourth, to increase the understanding of the micronutrient data (both dietary and biomarker-based), they were associated with socio-demographic characteristics and work-related factors through regression modelling. Outliers were identified as values that were more than three standard deviations away from the mean, and these were removed from the dataset. Stata^®^ version 15 (StataCorp, College Station, Texas, USA) was used for data analysis.

## 3. Results

### 3.1. Sociodemographic Characteristics of the E-Waste Recyclers and Controls

The mean ± SD age of the e-waste recyclers and the controls assessed was 25.4 ± 6.3 and 32.3 ± 10.2 years, respectively. BMI was significantly higher in the controls (23.9 kg/m^2^) than e-waste recyclers (21.8 kg/m^2^) but was within a healthy body weight range for both groups (Table 1). Majority of the e-waste recyclers (92%) and controls (84.3%) were Muslims (Table 1). More than half (55%) of e-waste recyclers were married, compared to (58.8%) of the controls who were single. Controls have had better education than e-waste recyclers; 52.3% of controls had completed Senior Secondary School, compared to just a third of the e-waste recyclers (32.3%) who reported to have completed Junior High School. Two thirds (63.6%) of e-waste recyclers earned between 20 and 100 Ghanaian cedis daily, and a 24.2% earned less than 20 Ghanaian cedis per day (~$4); and only 4% earned more than 200 Ghanaian cedis (~$35) per day. Furthermore, daily income earned was neither associated with educational status (χ^2^ = 4.42, *p* = 0.98) nor job task (χ^2^ = 6.73, *p* = 0.35) undertaken at the e-waste recycling site. More than half of the e-waste recyclers (55.2%) slept on-site, while the rest lived off-site either within 1 km of Agbogbloshie (36.5%) or more than 1 km away (8.3%). The recyclers reported to work approximately 9 h per day for about 10 years. Finally, 36% of the e-waste recyclers were highly exposed to biomass burning than the controls (29%).

### 3.2. Dietary Micronutrient Intake (Based on Food Models) of the E-Waste Recyclers and the Controls

Dietary micronutrient intake is summarized in Table 2. The dietary intake of Fe and Zn was significantly higher among the e-waste recyclers than controls, whereas Mg intake was higher among the controls (Table 2). For other micronutrients (Ca, Se, Fe, Mg, Zn, and Cu) assessed, there were no differences in consumption between e-waste recyclers and controls. For e-waste recyclers, the mean dietary intake of Ca, Se, Mg, Zn, and Cu was highest among those who were primarily involved in dismantling e-waste. In contrast, recyclers who were primarily involved in open-air burning of e-waste to retrieve valuable parts, especially Cu did not consume diets with adequate amounts of Ca, Zn, Cu, and Fe. Overall, nearly all the e-waste recyclers (96%) and controls (98%) met the RDA for Fe in this study. However, only a few of the e-waste recyclers (~10%) met the RDA for Ca, Mg, Se, and Cu (Appendix A).

### 3.3. Micronutrient Biomarker Levels (Descriptive Summary)

In general, both the blood and urine datasets were deemed fit for analyses in terms of our review of quality control parameters. The average recovery (accuracy) of elements from the blood standard reference materials (SRM) used was 94% of the expected value for Se, Cu, and Zn (Appendix A). However, the mean analytical precision of all blood elements was 9%. In urine, the mean recovery (accuracy) was between 63% and 123% for Cu, Zn, and Se. Analytical precision for the urinary micronutrient measured was 15%. The accuracy of Ca, Fe, and Mg in blood and urine could not be calculated as the reference materials used did not include guidance values for these.

For whole blood and urine samples analyzed, the mean concentrations obtained for Ca, Mg, Fe, Zn, Se, and Cu among the e-waste recyclers and the controls are reported in Table 3. We further tabulated these biomarker values per job tasks performed (Table 3).

Due to the unavailability of suitable biomarker reference levels for Ghana (or Africa), the mean micronutrient levels of Fe, Zn, Cu, and Se in blood and urine were compared to values proposed by Iyengar and Woittiez et al. [41] in their effort to establish “normal values” for elements in biological samples. However, blood and urinary levels of Ca and Mg were compared to reference ranges established by Alimonti et al. [42]. In all, median levels of Ca, Zn and Se measured in whole blood of both e-waste recyclers and controls fell within these reference ranges. The estimated levels of the micronutrients in urine were within the normal range in both the e-waste recyclers and controls.

The median blood levels of Mg, Se, and Fe were significantly lower among the e-waste recyclers than controls (*p <* 0.05). Furthermore, among the e-waste recycler groups, median levels of Ca, Mg, Zn, and Se in blood were considerably highest among those who self-reported that they primarily collect/sort e-waste, followed by the dismantlers and then the burners. Significant differences were found in urinary levels of Ca, Se, and Fe among the e-waste recyclers and controls (*p* < 0.01). While urinary Ca and Se levels were higher among the controls, increased amounts of Cu were excreted among the e-waste recyclers (*p <* 0.01). Urinary Zn and Cu levels differed among the e-waste recycler groups (*p* < 0.01). While Zn was particularly high in urine of the collectors/sorters, Cu was highly removed in urine of the burners.

### 3.4. Relationship between Dietary Micronutrient Intake and Micronutrient Levels in the Blood and Urine of E-Waste Recyclers

The correlations between dietary micronutrient intake and levels of micronutrient in the blood and urine of e-waste recyclers are presented in Table 4. Between dietary intake values and blood levels, few of these relationships were significantly positive (5 out of 36; Mg-Fe, Ca-Zn, Mg-Zn, Mg-Se, and Mg-Mg), whereas for urine only one was positively significantly associated (Mg-Cu). Further, ~50% of the micronutrients were found to be negatively correlated.

### 3.5. Association of Sociodemographic Characteristics and Other Work-Related Factors with Dietary Micronutrient Intake among E-Waste Recyclers

We further evaluated micronutrient-intake associations with daily work duration, age, daily income earned, educational status, exposure to biomass burning, cigarette smoking, alcohol intake, BMI, and recycler job task (Table 5). Factors such as hours spent recycling e-waste and daily income earned were associated with dietary Ca and Fe consumption. BMI was also positively associated with the consumption of foods rich in Mg, Cu, and Se. Most factors, such as age, job task undertaken, cigarette smoking, exposure to biomass burning and alcohol intake, were not associated with the dietary intake of micronutrient-rich diets.

Exposure to biomass burning was associated with the lower levels of Mg, Zn and Fe measured in the blood of e-waste recyclers (Table 5). Moreover, increased duration of work at the recycling site was inversely proportional to Mg levels in the blood of e-waste recyclers. The nature of e-waste job-task performed significantly influenced blood levels of Ca, Mg, Zn, and Cu, with collectors/sorters having the highest. However, factors such as daily work duration, age, BMI, cigarette smoking, and alcohol intake did not influence the blood micronutrient levels.

We also investigated the associations between job-related factors with urinary micronutrient excretion (Table 5). Significant relationships were found between years of recycling e-waste and urinary levels of Ca (β = 0.08; 95% CI: −0.01, 0.15, *p* = 0.02) and Cu (β = 0.09; 95% CI: 0.04, 0.13, *p* = 0.004). Urinary levels of Cu were also lower with increasing age (β = −0.07; 95% CI: −0.11, 0.03, *p* = 0.004). Factors such as exposure to biomass burning, alcohol intake, BMI, primary job task undertaken, and the number of hours (per day) spent recycling e-waste did not have any significant influence on urinary levels of the micronutrients measured.

## 4. Discussion

Recycling of electronic waste (e-waste) is growing worldwide [3,4]. It is largely an informal sector activity situated in low- and middle-income countries thus drawing from populations who are at the lower end of the economic ladder, and whose dietary intake is likely to not have the full complement of necessary or essential micronutrients [5]. Despite the growing concerns over health risks in these communities, no study to our knowledge has assessed the nutritional status of e-waste recyclers. Such an assessment is needed to increase understanding of the health of e-waste recyclers, and to better interpret environmentally related health risks due to exposure to toxicants, which has been the primary focus of most studies. To address this knowledge gap, our study is one of the first to estimate in detail the micronutrient status of e-waste recyclers, using both dietary survey information and biomarker data. We measured levels of micronutrients (Cu, Fe, Se, and Zn) previously and similarly assessed by Srigboh et al. [6] as well as Ca and Mg in whole blood and urine of e-waste recyclers at Agbogbloshie, Ghana. Subsequently, the levels of these micronutrients were compared to reference values proposed by Iyengar and Woittiez et al. [41] and Alimonti et al. [42] since there is no national, or continent specific reference values for micronutrients. Overall, among the e-waste recyclers, the blood levels of Ca, Se, and Zn reported here were within the proposed reference ranges. Meanwhile, among the controls, all the micronutrients measured in both the blood and urine fell within the reference ranges. There were differences in micronutrient intake or biomarker levels according to variables noted in our a priori objectives and these are discussed below.

### 4.1. Dietary Micronutrient Intake (Based on Estimates Using Food Models) of E-Waste Recyclers and Controls

Generally, dietary intake of Fe and Zn was adequate among e-waste recyclers probably owing to their regular intake of traditional green leafy soups. However, there was a high probability of inadequate dietary intake of Mg, Ca, Se, and Cu-rich diets in both e-waste recyclers and controls when compared to the RDA proposed by WHO. Particularly, the mean levels of Ca, Se, and Mg estimated from the diet were lower among the e-waste recyclers, while the mean level of Zn intake was lower among the controls. These results are consistent with similar outcomes in Malawi [43] and South Africa [44] where Ca and Se intake were inadequate among adult males (general population). Based on the inadequate intake of micronutrient-rich diets among the e-waste recyclers and controls, these findings seem to suggest micronutrient deficiency as a common public health issue among healthy males in sub-Saharan Africa. Studies have revealed micronutrient deficiencies in long-term heat-exposed steel recyclers as possibly due to huge losses through sweat and its further association with reduced appetite [45,46]. E-waste recyclers generally work in the open, and therefore tend to be exposed to vagaries of the weather. In particular, burners are highly exposed to heat, the dietary estimation using food models showed they least consumed Zn-rich foods believed to alleviate heat stress. Other studies have also predicted deficiencies in micronutrients (such as Ca, Cu, Mg, and Se) to be associated with increased exposure to toxic metals and other pollutants released during the informal processing of e-waste [15,45,46]. This suggests that in populations such as the informal sector of e-waste recyclers where toxicant exposures seem to be apparent, a public health strategy targeted at adequately increasing dietary intake of micronutrient-rich diets is critical.

### 4.2. Micronutrient Levels in Blood and Urine of E-Waste Recyclers and Controls

The analytical quality of the blood and urine measurements was good, and even though we did not have a reference material to gauge the accuracy of Ca, Fe, and Mg measurements, we do note that the values found here fall within normal reference ranges thus giving us some confidence in their validity. Blood levels of Ca, Mg, Se, and Fe were significantly lower among e-waste recyclers than the controls, with mean Mg and Fe levels of the e-waste recyclers falling below the reference ranges. The current study detected higher blood levels of Cu and Zn and lower blood levels of Fe and Se among e-waste recyclers in comparison with Srigboh et al. [6] who previously conducted a study in the same site. Furthermore, the blood level of Fe of e-waste recyclers was below the reference range used. The lower blood Fe levels of e-waste recyclers may not necessarily reflect the overall Fe status, as judging only by the low blood Fe levels may seem to indicate Fe deficiency, even when their Fe dietary intake met the RDA. Such as conclusion may need supported by other blood related biomarkers such as hemoglobin, total iron-binding capacity, transferrin receptor and ferritin, which were not analyzed in this study. Poor blood Ca and Mg levels of e-waste recyclers may be attributed to a higher intake of micronutrients such as Cu and Fe from their diet, which is believed to inhibit the absorption of Ca and Mg [47,48]. For instance, at the e-waste scrapyard, recyclers have a long-standing habit of consuming a lot of tea and other herbs believed to contain Cu, Zn, and Fe as well as organic acid, which are known to inhibit the absorption of Ca and Mg [49,50]. In addition, the lower blood levels of Ca, Mg, Se, and Fe among the e-waste recyclers may be related to their varied dietary habits and lower income status compared to the control population. Furthermore, among e-waste recycler groups, the lowest blood Ca levels were detected among the burners, who are highly exposed to heat, which mostly predisposes them to dehydration. In a bid to hydrate themselves, tend to drink a lot of carbonated drinks (mostly vended onsite) that contain phosphoric acid, believed to inhibit Ca and Mg absorption [51,52].

The estimation of 24-hr mineral excretion is known to give a more representative overview of micronutrient status. The median urinary levels of all the micronutrients analyzed fell within the reference ranges, indicating adequate micronutrient retention. When we further compared the micronutrient urinary levels detected in this study to the findings of Srigboh et al. [6], our urinary Zn and Cu levels among the current e-waste recyclers assessed were higher, implying a growing micronutrient excretion rate over time. In addition, we found higher urinary levels of Mg, Zn, Cu, and Fe among the e-waste recyclers than the controls. Specifically, between the recycler groups, the amount of Zn excreted was highest among the collectors/sorters. Given that excessive exposure to heat may lead to negative mineral balance and increased mineral excretion via urine [53,54], the high Cu levels excreted among burners are possible as one of the key tasks they undertake is to burn plastic off wires to recover Cu [6]. These higher excretion rates of Mg, Zn, Cu, and Fe among the e-waste recyclers may also be attributable to their higher exposure to toxic metals that are believed to interact, displace and remove micronutrients from the body [10,12,55].

### 4.3. Relationship between Dietary Micronutrient Intake and Micronutrient Levels in the Blood and Urine of the E-Waste Recyclers

The correlations between dietary Ca, Mg, Fe, Cu, Se, and Zn and the levels of these micronutrients in whole blood and urine of e-waste recyclers were computed. Contrary to our findings, Hunt and Beiseigel et al. [56] found no relationship between dietary Ca intake and Zn levels in the blood. However, Wood and Zheng et al. [57] found an inverse relationship between dietary Ca intake and blood Zn levels. Based on the above contradictory findings, the relationships between these micronutrients in different body fluids remains unclear. We further found an inverse correlation between dietary Mg intake and urinary Cu levels among the e-waste recyclers. This finding supports the results of Bulat et al. [58], who reported that Mg supplementation had profound effect on Cu status in cadmium-exposed animals. We also found a direct relationship between Mg intake from diet and Fe levels in the blood. Although some studies have attributed Fe deficiency anemia to excessive intake of Mg-rich diets [59], other studies report otherwise [60,61]. These descriptive correlation data may form the basis for further studies, and also support the notion of examining multiple elements in studies given that recyclers are exposed to complex mixtures.

### 4.4. The Relationship of Selected Work-Related Factors and Sociodemographic Characteristics with Dietary Micronutrient Intake and Micronutrient Levels in Blood and Urine of E-Waste Recyclers

Our results suggest that the e-waste recyclers’ Ca and Fe intake from their diet improved significantly with increasing daily income earned as well as number of hours spent recycling e-waste. This may be attributed to the fact that Ca and Fe containing foods are relatively expensive in urban areas [62], thus people with higher income status may afford such foods. Similarly, a systematic review confirmed that people who earned higher incomes were more likely to consume Ca and Fe-rich foods such as lean meats, whole grains, fish, and low-fat dairy products [63,64]. Furthermore, an increase in BMI was associated with an increase in Mg, Cu, and Se intake among the e-waste recyclers. In contrast, other studies have identified negative correlations between BMI and dietary intake of Se [65] and Mg [66]. Meanwhile, some other studies have found no differences between BMI and dietary intake of Mg, Cu, and Se [67,68,69]. These contrasts may be explained by the fact that our study did not consider the intake of dietary supplements; therefore, individual micronutrient intake may be underestimated, as supplement users are more likely to meet their daily requirements. Largely, factors such as e-waste job task undertaken, cigarette smoking, exposure to biomass burning, and alcohol intake were not associated with the micronutrients measured in this study. 

Generally, information about the effect of cigarette smoking on micronutrient status is scarce, except for Se and Cu. Our study found no significant relationship between cigarette smoking and micronutrient levels in blood. However, results from past studies have revealed that there is a significantly lower level of micronutrients in blood of cigarette smokers [70,71,72]. In addition, other studies have also found an inverse relationship between serum Zn and Cu levels and cigarette smoking; however, these relationships were left unexplained [73,74,75]. Drawing parallels from these findings, the effects of cigarette smoking on micronutrient status may be dependent on the type, quantity as well as how smoking is done. Our study did not consider all these factors; thus, the possible lack of relationship observed. Moreover, we found that exposure to biomass burning significantly reduced blood levels of Fe, Mg, and Zn among the e-waste recyclers. This negative association between exposure to biomass burning and blood Fe levels, for instance, may probably be the reason for the low blood Fe levels measured among e-waste recyclers, which may consequently serve as a risk factor for anemia. Furthermore, we observed that prolonged stay and recycling of e-waste over time might be associated with reduction in Mg absorption in the blood of e-waste recyclers. These lower blood Mg levels may further be linked to increased exposure to toxicants such as Pb, As, and Cd, which are released during informal e-waste recycling activities. The differences in blood levels of Ca, Mg, Zn, and Cu found among the e-waste recycler groups may be attributed to their job task and degree of exposure to toxicant as evident in Srigboh et al. [6] findings.

Assessing the association between work-related factors, sociodemographic characteristics and urinary excretion of micronutrients, we found that prolong years of e-waste recycling directly influenced the amount of Ca and Cu excreted. Toxic metals such as Cd and Pb have slower excretion rates from the body, yet long half-lives (e.g., the half-life of Pb is 27 years in cortical bone and 16 years in cancellous bone, whereas Cd has a half-life of 10–30 years) [76]. In view of these, working at the e-waste site for at least 10 years may allow already bio-accumulated metals to interact, displace and excrete micronutrients such as Ca and Cu through the urine in humans. In addition, we found that as age increases, the amount of Cu excreted in urine reduced among the e-waste recyclers.

Given the overview of the micronutrient status of the e-waste recyclers and controls, the employment of food-based approaches like supplementation and fortification strategies may improve micronutrient status; however, these approaches also have some limitations, considering the potential interactions that may occur when certain supplements are consumed together and in varied quantities. For instance, high-dose Fe supplements can reduce the absorption of Zn, whereas high amounts of Zn may inhibit the absorption of Cu [77,78]. Therefore, in order to meet micronutrient needs and optimize health, clinicians like dieticians and nutritionists should educate the public about healthy dietary patterns, as well as safe and appropriate selection and use of micronutrient supplements.

This study has several strengths. It is well known that dietary intake surveys are mostly challenged by the risk of under- and/or over-reporting. Our study therefore employed biomarker analysis of micronutrients in biological samples (blood and urine) to help support the survey results. This approach may provide a more accurate and holistic understanding of micronutrient status. This study is arguably the largest examination of micronutrient status among e-waste recyclers, and is situated in one of the best-studied e-waste sites worldwide and as such benefits from a range of other data and knowledge available. However, in future studies, the use of repeated measurements of micronutrient biomarkers will be required to better characterize the micronutrient status of the recyclers and their effects on health outcomes. Further, we asked questions about job activities and present a simple interpretation here (three general categories) though other studies have shown that e-waste recyclers tend to perform many tasks on-site [29]. We tried to put the findings into context by comparing values with “background populations” but there is a lack of understanding of such values from Africa, especially from vulnerable recycler populations based in low- and middle-income countries.

## 5. Conclusions

In conclusion, our findings highlight inadequate intake of Ca, Mg, and Cu rich-diets among e-waste recyclers and controls. Clearly, a coordinated multi-micronutrient program is needed to combat the co-existing micronutrient deficiencies in not only e-waste recyclers but also the general population; at least in those with poor income status in the region based on a review of the data from the controls. We also found that median urinary levels of all the micronutrients analyzed fell within reference ranges as established by Iyengar and Woittiez in 1988. Although not conclusive, the lower blood levels of Fe among e-waste recyclers may indicate Fe deficiency. We further identified that exposure to biomass burning was associated with lower blood levels of Fe, Mg and Zn among the e-waste recyclers, which may consequently serve as a risk for developing anemia. Prolong stay and recycling of e-waste coupled with exposure to toxic metals that have a half-life as long as 10 years may increase the risk of micronutrient excretion among e-waste recyclers than controls. This may serve as a risk of micronutrient deficiency among e-waste recyclers. Furthermore, the inadequate intake of Zn-rich foods among the burners, coupled with their increased risk of heat exposure, negatively influences their Zn status.

Specific RDAs, as well as biomarker reference levels tailored to toxicant-exposed groups, remain absent, making the comparisons unparalleled. We, therefore, recommend the need to obtain micronutrient biomarker reference levels for toxicant-exposed groups, such as the e-waste recyclers. Furthermore, country-specific reference levels for biomarkers need to be set for ideal comparisons. We also recommend, “omics” technology such as non-targeted mass spectrometry-based metabolomics would provide much more information about the effect of chronic micronutrient deficiencies or heavy metal exposure [79]. In addition, studies should be conducted to identify appropriate RDAs for toxicant-exposed groups, as they may have a higher requirement than the allowances for healthy people. We also recommend nutrition program interventions, including micronutrient supplementation, education, and health monitoring for e-waste recyclers who are believed to be at risk of malnutrition. Beyond that, tailored nutrition-related dialogues are required to educate informal e-waste recyclers on the ideal intake of specific nutrients of concern and their impact on nutritional status in order to improve the health of people in a highly polluted environment.

## Figures and Tables

**Table 1 ijerph-17-09575-t001:** Socio-demographic characteristics and anthropometric measures of e-waste recyclers and controls.

Demographics	Total N	E-Waste Recyclers *n* (%)	Controls *n* (%)	X^2^	*p*-Value
**Socio-demographic characteristics**					
Marital Status	150			4.43	0.06
Single		44 (44.4)	31 (60.8)		
Married		55 (55.6)	20 (39.2)		
Daily Income	149			5.12	0.16
≤GHS 20		24 (24.2)	9 (18.0)		
GHS 21–100		63 (63.6)	30 (60.0)		
GHS 101–200		8 (8.1)	4 (8.0)		
GHS > 200		4(4.0)	7 (14.0)		
Education	145			23.82	<0.01
None		25 (25.2)	6 (13.0)		
Primary		26 (26.3)	4 (8.7)		
Middle/JSS		32 (32.3)	12 (26.1)		
Secondary/SSS & Higher		16 (16.2)	24 (52.3)		
Religion	150			3.45	0.18
Muslim		92 (92.9)	43 (84.3)		
Christian		5 (5.1)	7 (13.7)		
Others		2 (2)	1 (2)		
Smoking	36	27 (27.8)	6 (12.4)	4.52	0.03
Alcohol intake	26	17 (17.0)	9 (17.7)	0.01	0.92
E-waste Job-task	100			NA	NA
Burners		32 (32)	NA		
Dismantlers		49 (49)	NA		
Collectors/Sorters		19 (19)	NA		
		**Anthropometric Measures**			
		**Mean ± SD**	**Mean ± SD**		***p*-value**
Weight (kg)		63.4 ± 9.5	71.6 ±12.6		<0.01
Height (m)		1.71 ± 0.1	1.73 ± 0.1		0.07
BMI (kg/m^3^)		21.8 ± 2.7	23.9 ± 3.5		<0.01

Abbreviation: SD; Standard Deviation; NA: Not applicable.

**Table 2 ijerph-17-09575-t002:** Dietary micronutrient intake (based on food models) of participants.

	**Dietary Micronutrient Intake of E-Waste Recyclers and Controls**							
**Dietary Micronutrient** **Intake (mg)**		**E-Waste Recyclers (n = 100)**		**Controls (n = 51)**				
	**RDA (mg)**	**Mean ± SD**	**Median (IQR)**	**Mean ± SD**	**Median (IQR)**	***p*-Value**		
Ca	1000	534.4 ± 352.3	449 (375.1)	557.0 ± 369.6	497 (321)	0.47		
Mg	350	45.7 ± 44.2	33.3 (38.1)	84.7 ± 67.3	59.6 (86.8)	<0.01		
Se	55	22.8 ± 18.0	17.9 (20.4)	32.2 ± 39.9	19.8 (36.2)	0.50		
Zn	11	11.3 ± 4.1	10.6 (4.6)	9.7 ± 4.7	8.7 (5.9)	0.01		
Cu	2	1.1 ± 0.6	1.0 (0.6)	1.0 ± 0.7	1.1 (0.6)	0.12		
Fe	8	26.5 ± 13.0	24.2 (15.9)	22.4 ± 10.5	24.8 (14.4)	0.04		
	**Dietary Micronutrient Intake Per E-Waste Recycler-Group**							
**Dietary Micronutrient** **Intake (mg)**		**Burner (n = 32)**		**Dismantler (n = 49)**		**Collector/Sorter (n = 19)**		
	**RDA (mg)**	**Mean ± SD**	**Median (IQR)**	**Mean ± SD**	**Median (IQR)**	**Mean ± SD**	**Median (IQR)**	***p*-Value**
Ca	1000	453.3 ± 164.8	418.3 (202.5)	582.8 ± 439.1	468.5 (400.6)	546.3 ± 321.5	534.5 (376)	0.67
Mg	350	46.1 ± 38.4	39.9 (50.6)	49.7 ± 52.5	34.3 (32.7)	34.1 ± 24.7	26.9 (38.3)	0.48
Se	55	21.8 ± 18.3	16.3 (21.4)	24.3 ± 18.7	19.0 (18.1)	20.9 ± 16.3	16.4 (24.6)	0.57
Zn	11	10.5 ± 4.2	10.3 (5.7)	11.9 ± 4.3	11.4 (5.1)	11.2 ± 3.3	10.6 (4.3)	0.61
Cu	2	1.0 ± 0.5	1.0 (0.7)	1.2 ± 0.7	1.1 (0.6)	1.1 ± 0.5	1.0 (0.8)	0.79
Fe	8	23.9 ± 12.9	22.7 (13.3)	27.7 ± 13.1	24.8 (14.4)	27.8 ± 12.9	25.4 (18.6)	0.35

Abbreviations; SD: standard deviation; IQR: interquartile range; Ca: Calcium; Mg: Magnesium; Fe: Iron; Se: Selenium; Cu: Copper; Zn: Zinc. *p*-value estimate from the Wilcoxon rank sum test (e-waste recyclers *vrs* controls) whilst *p*-value estimate per median differences in dietary intake between recyclers were derived from Kruskal–Wallis test. The recycler-groups were determined based on self-reported data.

**Table 3 ijerph-17-09575-t003:** Micronutrient levels in blood and urine of participants.

	**Blood and Urinary Micronutrient Levels Analyzed among E-Waste Recyclers and Controls**							
**Micronutrient Levels (µg/L)**	**Reference Range (µg/L)**	**E-Waste Recyclers (n = 100)**		**Controls (n = 51)**				
		**Mean ± SD**	**Median (IQR)**	**Mean ± SD**	**Median (IQR)**	***p*-Value**		
**Whole blood levels**								
Ca	59,028–72,193	61,992.1 ± 13,610.6	63,155.9 (16,083.2)	65,851.8 ± 9270.3	65,629.4 (8835.7)	0.07		
Mg	36,951–43,276	33,717.2 ± 5694.1	34,581.0 (6703.3)	38,363.8 ± 4653.1	38,560.7 (6481.2)	<0.01		
Se	58–234	152.9 ± 39.8	145.9 (56.9)	194.9 ± 44.1	193.1 (49.4)	<0.01		
Zn	4837–7980	7879.3 ± 2782.3	7242.9 (3267.6)	8355.2 ± 2466.9	8057.9 (2043.2)	0.09		
Cu	683–1036	1107.9 ± 222.6	1110.9 (233.5)	1143.0 ± 234.1	1107.7 (259.6)	0.72		
Fe	390,000–550,000	370,697.2 ± 68,181.2	381,307.2 (77,495.4)	404,169.7 ± 65,185.7	427,274.6 (88,040.3)	<0.01		
**Urinary levels**								
Ca	67,000–200,000	66,419.2 ± 75,335.6	40,243.9 (62,094.9)	90,814.3 ± 69,028.3	71,103.2 (96,152.5)	<0.01		
Mg	15,000–120,000	82,220.5 ± 55,215	71,291.8 (72,978.9)	82,000.5 ± 56,931.5	76,954.6 (63,733.0)	0.98		
Se	7–160	29.7 ± 18.8	26.7 (22.3)	44.5 ± 25.8	39.9 (31.1)	<0.01		
Zn	700–2500	3718.7 ± 7942.6	1044.4 (5022.0)	1561.8 ± 1294.2	1253.0 (1195.7)	0.89		
Cu	12–80	63.7 ± 53.8	38.1 (76.6)	110.5 ± 567.8	25.6 (16.0)	<0.01		
Fe	1.2–600	181.4 ± 351.8	95.4 (79.8)	114.4 ± 97.8	85.2 (52.3)	0.08		
	**Blood and Urinary Micronutrient Levels Analyzed between E-Waste Recycler-Groups**							
**Micronutrient Levels (µg/L)**	**Reference Range (µg/L)**	**Burner (n = 32)**		**Dismantler (n = 49)**		**Collector/Sorter (n = 19)**		
		**Mean ± SD**	**Median (IQR)**	**Mean ± SD**	**Median (IQR)**	**Mean ± SD**	**Median (IQR)**	***p*-Value**
**Whole blood levels**								
Ca	59,028–72,193	55,577.0 ± 13,646.0	55,429.0 (18,578.1)	62,995.2 ± 11,426.7	63,719.6 (12,923.5)	70,209.5 ± 14,240.9	68,755.8 (11,117.3)	<0.01
Mg	36,951–43,276	30,991.6 ± 6462.2	32,633.0 (7851.5)	34,903.6 ± 5403.7	35,413.4 (6951.0)	35,248.3 ± 5132.2	36,003.3 (6693.6)	0.02
Se	58–234	135.1± 34.1	129.5 (33.8)	158.5 ± 41.8	159.7 (63.3)	168.6 ± 33.9	162.4 (51.5)	<0.01
Zn	4837–7980	6496.3 ± 2114.9	6667.9 (2356.9)	8105.2 ± 2594.0	7571.8 (2747.2)	9626.1 ± 3189.2	10,069 (4126.8)	<0.01
Cu	683–1036	1038.4 ± 249.3	1028.6 (324.4)	1126.5 ± 218.5	1107.8 (217.7)	1173.6 ± 155.2	1131.6 (198.1)	0.12
Fe	390,000–550,000	350,216.2 ± 78,098.1	368,696.2 (120,304.8)	382,397.1 ± 58,722.5	384,963.9 (67,423.7)	375,166.5 ± 68,887.8	389,211.4 (75,233.2)	0.19
**Urinary Levels**								
Ca	67,000–200,000	61,621.6 ± 73,714.8	34,674.6 (67,557.2)	58,091.7 ± 47,212.3	53,992.3 (60,345.3)	95,099.1 ± 120,164.4	40,548.7 (79,110.5)	0.44
Mg	15,000–120,000	85,002.8 ± 62,528.2	66,585.8 (81,500.2)	78,503.6 ± 53,177.6	71,412.0 (67,174.9)	86,729.0 ± 48,965.0	86,529.7 (59,442.0)	0.70
Se	7–160	26.3 ± 20.2	20.7 (19.9)	32.3 ± 19.2	27.1 (22.9)	31.6 ± 15.1	32.3 (23.3)	0.15
Zn	700–2500	1548.1 ± 2012.8	784.5 (769.1)	5368.0 ± 11,138.6	1047.6 (5022)	3468.2 ± 2729.4	4597.4 (5473.2)	<0.01
Cu	12–80	87.7 ± 57.6	97.6 (87.3)	48.3 ± 57.6	87.3 (82.6)	61.2 ± 64.8	33.8 (43.0)	<0.01
Fe	1.2–600	108.9 ± 69.4	87.3 (82.6)	108.9 ± 69.4	100.4 (68.9)	426.5 ± 744.7	100.4 (153.6)	0.19

Abbreviations: Ca: Calcium; Mg: Magnesium; Fe: Iron; Se: Selenium; Cu: Copper; Zn: Zinc. Sources of reference ranges: Se, Fe, Zn, Cu -Iyengar and Woittiez et al. [41]; whiles Ca and Mg were from-Alimonti et al. [42] in blood and urine.

**Table 4 ijerph-17-09575-t004:** Correlation coefficients between dietary (D) micronutrient intake (top row) and micronutrient levels in blood (B) and urine (U) of e-waste recyclers (first column).

Biomarkers (µg/L)	D-Ca (mg)	D-Mg (mg)	D-Fe (mg)	D-Se (mcg)	D-Cu (mg)	D-Zn (mg)
B-Ca	0.15	0.17	0.06	0.11	−0.04	−0.002
B-Mg	0.11	**0.27 ****	0.04	0.09	−0.03	0.002
B-Fe	0.17	**0.17 ***	0.10	0.10	0.001	0.06
B-Cu	−0.02	0.17	0.04	0.02	−0.04	0.001
B-Zn	**0.29 ****	**0.18 ***	0.11	0.13	−0.03	0.08
B-Se	0.16	**0.21 ***	0.07	0.07	−0.06	0.002
U-Ca	0.04	−0.09	−0.06	0.07	**0.21 ****	−0.10
U-Mg	−0.003	−0.10	−0.07	0.03	−0.06	−0.10
U-Fe	0.11	−0.04	0.07	0.01	−0.02	0.09
U-Cu	−0.05	**−0.21 ****	−0.09	0.05	0.09	−0.11
U-Zn	0.07	0.09	0.02	0.12	−0.04	−0.01
U-Se	0.06	−0.11	−0.12	0.12	0.01	−0.09

Abbreviations: D-Dietary; U-Urinary; B-Blood; *p*-value notation: * *p* < 0.05, *** p* < 0.01; Bolded numbers indicate correlations that are of statistical significance.

**Table 5 ijerph-17-09575-t005:** Relationship of socio-demographic characteristics and job-related factors with dietary micronutrient intake (mg) and micronutrient levels in blood **(µg/L)** and urine **(µg/L)** of e-waste recyclers.

**Variables**	**D-Ca (mg)** **β [95% CI]**	**D-Mg (mg)** **β [95% CI]**	**D-Fe (mg)** **β [95% CI]**	**D-Zn (mg)** **β [95% CI]**	**D-Cu (mg)** **β [95% CI]**	**D-Se (mcg)** **β [95% CI]**
Daily duration (hours) of e-waste work activity	**−0.05 *** [−0.09, −0.01]	0.05 [−0.02, 0.12]	**−0.05 *** [−0.08, −0.01]	−0.01 [−0.05, 0.02]	0.07 [−0.02, 0.16]	0.08 [−0.03, 0.18]
Educational Status	−0.06 [−0.16, 0.04]	**−0.19 *** [−0.36, −0.02]	−0.01 [−0.10, 0.09]	−0.02 [−0.11, 0.07]	−0.10 [−0.31, 0.11]	−0.07 [−0.33, 0.19]
Daily Income earned	**0.11 *** [0.01, 0.21]	−0.05 [−0.21, 0.12]	**0.15 *** [0.01, 0.30]	0.10 [−0.04, 0.24]	−0.16 [−0.50, 0.17]	−0.07 [−0.47, 0.34]
Age (years)	0.02 [0.002, 0.04]	0.03 [−0.002, 0.07]	0.02 [−0.003, 0.03]	0.001 [−0.02, 0.02]	−0.0003 [−0.04, 0.04]	−0.03 [−0.08, 0.02]
BMI (kg/m^3^)	−0.01 [−0.05, 0.04]	**0.08 *** [−0.002, 0.16]	−0.01 [−0.05, 0.04]	0.03 [−0.01, 0.07]	**0.11 *** [0.02, 0.21]	**0.15 *** [0.03, 0.26]
Cigarette smoking	0.06 [−0.19, 0.30]	−0.20 [−0.62, 0.22]	−0.06 [−0.29, 0.17]	0.04 [−0.18, 0.26]	−0.01 [−0.53, 0.51]	−0.15 [−0.77, 0.48]
Biomass exposure	0.14 [−0.08, 0.36]	−0.06 [−0.46, 0.33]	0.13 [−0.07, 0.34]	0.06 [−0.14, 0.25]	0.20 [−0.26, 0.66]	−0.37 [−0.92, 0.19]
Alcohol intake	−0.07 [−0.37, 0.23]	−0.03 [−0.57, 0.51]	0.01 [−0.07, 0.09]	−0.02 [−0.09, 0.06]	0.01 [−0.17, 0.19]	−0.13 [−0.11, 0.36]
Job task	−0.06 [−0.23, 0.11]	−0.19 [−0.48, 0.10]	−0.01 [−0.17, 0.15]	0.09 [−0.07, 0.24]	0.01 [−0.35, 0.37]	0.17 [−0.26, 0.60]
**Variables**	**B-Ca (µg/L)** **β [95% CI]**	**B-Mg (µg/L)** **β [95% CI]**	**B-Fe (µg/L)** **β [95% CI]**	**B-Zn (µg/L)** **β [95% CI]**	**B-Cu(µg/L)** **β [95% CI]**	**B-Se (µg/L)** **β [95% CI]**
Daily duration (hours) of e-waste work activity	0.10 [−0.01, 0.03]	0.01 [−0.01, 0.03]	0.003 [−0.01, 0.02]	−0.001 [−0.03, 0.03]	0.002 [−0.01, 0.03]	−0.003 [−0.02, 0.02]
Years of performing e-waste activities (years)	−0.003 [−0.02, 0.01]	**−0.01 *** [−0.02, −0.001]	−0.01 [−0.02, 0.01]	−0.01 [−0.02, 0.01]	−0.01 [−0.02, 0.002]	−0.004 [−0.02, 0.01]
Cigarette smoking	−0.07 [0.20, 0.05]	−0.07 [−0.19, 0.04]	−0.05 [−0.17, 0.07]	−0.10 [−0.28, 0.08]	−0.07 [−0.19, 0.05]	−0.10 [−0.23, 0.03]
Biomass exposure	−0.01 [−0.11, 0.09]	**−0.10 *** [−0.20, 0.003]	**−0.15 **** [−0.25, −0.05]	**−0.24 **** [−0.40, −0.09]	−0.03 [−0.14, 0.06]	−0.01 [−0.12, 0.11]
Age (years)	−0.004 [−0.01, 0.01]	0.01 [−0.004, 0.02]	0.003 [−0.01, 0.01]	−0.002 [−0.02, 0.01]	0.002 [−0.01, 0.01]	0.01 [−0.01, 0.02]
BMI (kg/m^3^)	0.01 [−0.01, 0.03]	0.001 [−0.02, 0.02]	0.01 [−0.02, 0.03]	0.01 [−0.02, 0.04]	−0.001 [−0.02, 0.02]	0.01 [−0.01, 0.03]
Alcohol intake	−0.10 [−0.25, 0.05]	−0.06 [−0.20, 0.08]	−0.05 [−0.19, 0.10]	−0.07 [−0.28, 0.15]	−0.02 [−0.17, 0.12]	−0.05 [−0.21, 0.11]
Job task	**0.14 **** [0.06, 0.23]	**0.09 *** [0.01, 0.17]	0.03 [−0.05, 0.12]	**0.18 **** [0.05, 0.30]	**0.10 *** [0.01, 0.18]	0.09 [−0.003, 0.18]
**Variables**	**U-Ca (µg/L)** **β [95% CI]**	**U-Mg (µg/L)** **β [95% CI]**	**U-Fe (µg/L)** **β [95% CI]**	**U-Zn (µg/L)** **β [95% CI]**	**U-Cu (µg/L)** **β [95% CI]**	**U-Se (µg/L)** **β [95% CI]**
Daily duration (hours) of e-waste work activity	−0.07 [−0.18, 0.04]	−0.05 [−0.12, 0.02]	0.01 [−0.07, 0.09]	0.03 [−0.09, 0.16]	−0.01 [−0.08, 0.07]	−0.03 [−0.09, 0.02]
Years of performing e-waste activities (years)	0.08 * [−0.02, 0.15]	0.02 [−0.02, 0.06]	0.03 [−0.02, 0.08]	−0.02 [−0.09, 0.05]	**0.07 *** [0.02, 0.12]	−0.003 [−0.04, 0.03]
Cigarette smoking	**−0.77 *** [−1.42, −0.12]	**−0.60 *** [−1.00, 0.19]	0.02 [−0.36, 0.59]	**−0.85 *** [−1.58, −0.13]	0.16 [−0.30, 0.63]	−0.33 [−0.67, 0.003]
Biomass exposure	−0.06 [−0.62, 0.50]	−0.25 [−0.60, 0.10]	−0.20 [−0.61, 0.20]	0.01 [−0.61, 0.64]	0.06 [−0.33, 0.46]	0.05 [−0.25, 0.32]
Age (years)	−0.04 [−0.10, 0.02]	−0.02 [−0.05, 0.02]	−0.01 [−0.05, 0.03]	0.001 [−0.06, 0.07]	**−0.06 *** [−0.10, −0.02]	−0.02 [−0.05, 0.01]
BMI (kg/m^3^)	0.01 [−0.10, 0.02]	−0.02 [−0.10, 0.05]	0.01 [−0.07, 0.98]	−0.02 [−0.15, 0.11]	0.05 [−0.03, 0.13]	0.05 [−0.01, 0.11]
Alcohol intake	0.21 [−0.57, 1.00]	0.06 [−0.43, 0.55]	−0.31 [−0.88, 0.27]	0.06 [−0.82, 0.93]	0.07 [−0.49, 0.63]	0.29 [−0.12, 0.70]
Job task/recycler group	0.125 [−0.30, 0.61]	−0.01 [−0.29, 0.27]	0.25 [−0.08, 0.59]	0.41 [−0.09, 0.92]	−0.11 [−0.43, 0.21]	0.21 [0.03, 0.44]

Abbreviations: D-Ca: Dietary Calcium; D-Mg: Dietary Magnesium; D-Fe: Dietary Iron; D-Cu: Dietary Copper; D-Se: Dietary Selenium; D-Zn: Dietary Zinc; B-Ca: Blood Calcium; B-Mg: Blood Magnesium; D-Fe: Blood Iron; D-Cu: Blood Copper; D-Se: Blood Selenium; D-Zn: Blood Zinc; U-Ca: Urinary Calcium; U-Mg: Urinary Magnesium; U-Fe: Urinary Iron; U-Cu: Urinary Copper; U-Se: Urinary Selenium; U-Zn: Urinary Zinc; *p*-value notations: *p* < 0.05 *; *p* < 0.01 **.

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
