# Peer review of "Micronutrient Status of Electronic Waste Recyclers at Agbogbloshie, Ghana"

_ijerph, 2020, doi:10.3390/ijerph17249575_

Round 1

Reviewer 1 Report

  1. The abstract should contain the name of the location where the data collection was conducted (Agbogbloshie) in the methods description and not only later in the results.
  2. The introduction is we;;-written generally, and the flow of information is acceptable. Please make sure to un-bold lines 60 and 61.
  3. The methods: you have a sentence that is repeated twice, lines 73 and 74. Please remove it.
  4. I would prefer that the cohort used for the study is described in more details, mainly the aim of the cohort and the number of participants.
  5. Why the samples were not analyzed locally, and sent to Canada for analysis?
  6. Line 188 contains a repeated phrase (associated with).
  7. More information is needed about the control group, regarding their occupations status and possible similar exposures.
  8. I think the tables need some organization, especially their titles and footnotes. Also, the page setup within the tables needs to be fixed.

Reviewer 2 Report

The research assessed the micronutrient status of e-waste recyclers using dietary information (2-day 24-hour recall survey) and biomarker data (whole blood and urine) among 151 participants (100 e-waste recyclers and 51 controls from the Accra region, Ghana) to study the relationship between dietary micronutrient intake and the micronutrient levels in blood and urine, and to examine the associations between work-related factors and sociodemographic characteristics on dietary micronutrient intake, as well as the micronutrient levels in the blood and urine of the e-waste recyclers and controls.

I have some comments and suggestions for the authors:

The linear regression models mentioned in the abstract (line 20) and the part of results (lines 294 to 318), but the tables were put in the supplementary. I would like to suggest the supplementary tables 4a to 4c change to be part of result. It could put within lines 294 to 318, page 14.

In the table 1, the number of alcohol intake was 151. Is it a mistake?

The anthropometric measures in supplementary table 2 should be moved to Table 1 (page 6) combined.

Numbers of workers and controls should presented in the tables 2a, 2b, and 3a.

What were the statistical methods (tests) used in the tables 3a and 3b?

What was correlation used in the table 4? Pearson or Spearman?

The limitation of this study was small sample size, and the control were better on education (as well as socioeconomic level) and anthropometric measures. The authors should discuss these points affecting the study and resulting in the conclusion.

Author Response

Response to Reviewer 2 Comments

Point 1: The linear regression models mentioned in the abstract (line 20) and the part of results (lines 294 to 318), but the tables were put in the supplementary. I would like to suggest the supplementary tables 4a to 4c change to be part of result. It could put within lines 294 to 318, page 14.

Response 1: As suggested, we have moved the initial supplementary tables 4a to 4c into the main results section, and they are now specifically within lines 323 to 343, from pages 14 to 16 and re-numbered as table 5a, table 5b and table 5c.

Point 2: In the table 1, the number of alcohol intake was 151. Is it a mistake?

Response 2: Thank you for drawing our attention.  This number is a mistake and therefore the typo has been corrected in table 1, which now reads as “26” on page 7 to 8.

Point 3: The anthropometric measures in supplementary table 2 should be moved to Table 1 (page 6) combined.

Response 3: The anthropometric measures initially in supplementary table 2 has been merged with Table 1 on page 7 to 8 in the main results as suggested.

Point 4: Numbers of workers and controls should presented in the tables 2a, 2b, and 3a.

Response 4: We have now presented the number of e-waste recyclers and controls in tables 2a, 2b and 3a accordingly as suggested. These changes are found on pages 8 to 11.

Point 5: What were the statistical methods (tests) used in the tables 3a and 3b?

Response 5: The Wilcoxon rank-sum test was used to compare the median differences of micronutrients measured in blood and urine between the e-waste recyclers and controls (table 3a), whiles the Kruskal-Wallis test was used to examine the median differences in biomarker micronutrient levels between the e-waste recycler groups (table 3b). This explanation is already documented in lines 203 to 210 in the methods section and it reads:

The Wilcoxon rank-sum test was used to determine whether differences existed between the amounts of micronutrients consumed from diet by the e-waste recyclers and the controls”

Second, to compare the levels of median differences in micronutrients measured in the whole blood and urine between the e-waste recyclers and the controls, the Wilcoxon rank-sum test was used. We further used the Kruskal-Wallis test to examine the median differences in biomarker micronutrient levels measured between the e-waste recycler groups”.

Point 6: What was correlation used in the table 4? Pearson or Spearman?

Response 6: We used the Pearson’s correlation to assess the relationship between dietary intake and micronutrient levels in blood and urine. This was clearly documented in lines 210 and 212 of the methods section of the manuscript, which reads as:

“Third, relationships between dietary micronutrient intakes and micronutrient levels in blood and urine were gauged using Pearson’s correlation”.

Point 7: The limitation of this study was small sample size, and the control were better on education (as well as socioeconomic level) and anthropometric measures. The authors should discuss these points affecting the study and resulting in the conclusion.

Response 7: Yes, we agree with the reviewer that there were differences in the level of education and anthropometric measures between the controls and the e-waste recyclers. To address this selection bias, a multivariable regression analyses was further conducted to adjust for the differences in the aforementioned measures.

Reviewer 3 Report

The authors have conducted an investigation into the micronutrient status of e-waste recyclers and control in Agbogbloshie in central Accra, Ghana, which has important practical significance. The conclusion of the project has important reference value for the relevant administrative decisions of the local government. However, there are several elements of the manuscript that need further attention.

1.The environmental pollution caused by e-waste involves a variety of toxic heavy metals, plastics, brominated flame retardants, color matching agents, surface coatings and so on. In addition to the heavy metals described in the article, are there any materials about other heavy metals, such as mercury (Hg), lead (Pb), chromium (Cr), cadmium (Cd), etc., which influence the micronutrient status of exposure population and can induce a variety of adverse health effects in them?

2.The micronutrient includes mineral substance and vitamin, only six of mineral substance are determined in the research without vitamin. Actually, vitamin is one of the most important substance of the micronutrient status in human body, there should be more explanation for the study.

3.Keywords should be more concise.

4.The inclusion criteria of sample population only include two points, which is not sufficient in this study. At the same time, exclusion criteria of sample population should be displayed? For example, people with organic and chronic diseases, or people who are taking drugs, etc., are included?

5.Generally, the 3-day 24-hour retrospective method was used in dietary survey, in which, the survey time should be randomly selected from Monday to Sunday for there may be significant difference in the dietary quantity and quality between working days and rest days and so on.

6.The correlation coefficient between dietary micronutrient intake and micronutrient levels in blood and urine of e-waste recyclers are relative low, but their statistical difference was significant in table 4?

7.According to results of this paper, e-waste environmental pollution has little impact on the health of the population? The most important factors of insufficient micronutrient state of e-waste recyclers and general population is socioeconomic factor? 

Author Response

Response to Reviewer 3 Comments

POINT 1: The environmental pollution caused by e-waste involves a variety of toxic heavy metals, plastics, brominated flame retardants, color matching agents, surface coatings and so on. In addition to the heavy metals described in the article, are there any materials about other heavy metals, such as mercury (Hg), lead (Pb), chromium (Cr), cadmium (Cd), etc., which influence the micronutrient status of exposure population and can induce a variety of adverse health effects in them?

Response 1: Many thanks for your question. Most toxic metals like Pb, Cd, Cr and Hg mimic micronutrients in cellular processes; thus, more chemicals than micronutrients are rather absorbed and transported (Jan et al., 2015; Nordberg et al., 2007). For instance, mechanistically, Pb may mimic Ca by entering voltage-activated L-type Ca channels in place of Ca (Goyer, 1997; Nordberg et al., 2007). Again, Cd exhibits an inhibitory effect on Fe metabolism and absorption, which may cause decreases in haematocrit and hemoglobin levels (Horiguchi et al., 2011; Peraza et al., 1998). Consequently, the toxic metals interact with micronutrients in blood, causing the displacement and subsequent excretion of micronutrients in urine, (Bonner et al., 1981), faeces (Brzoska et al., 2000) and sweat (Sears et al., 2012) thus leading to unavailability of most micronutrients. These interactions may induce variety of adverse health effects spanning from anaemia to diabetes, hypertension and other non-communicable diseases. This detail has been inserted in lines 54 to 59, which reads:

“For instance, Pb may mimic Ca by entering voltage-activated L-type Ca channels in place of Ca , and Cd exhibits an inhibitory effect on Fe metabolism and absorption, which may cause decreases in haematocrit and hemoglobin levels (Horiguchi et al., 2011; Peraza et al., 1998). In addition, several toxic metals may interact with micronutrients in blood, causing the displacement and subsequent excretion of micronutrients in urine, (Bonner et al., 1981), faeces (Brzoska et al., 2000) and sweat (Sears et al., 2012)”.

Point 2: The micronutrient includes mineral substance and vitamin, only six of mineral substance are determined in the research without vitamin. Actually, vitamin is one of the most important substance of the micronutrient status in human body, there should be more explanation for the study.

Response 2: We agree with you that vitamins are one of the most important substances in assessing the micronutrient status in the human body. However, this study set out to mainly examine specifically minerals such as Calcium (Ca), Magnesium (Mg), Iron (Fe), Selenium (Se), Copper (Cu), Zinc (Zn) which have been related to the occurrence of metabolic disorders. As authors, we may consider assessing the micronutrients status of this population using a wider range of minerals as well as vitamins in future.

Point 3: Keywords should be more concise.

Response 3: We have reviewed the keywords and also used U.S. NCBI’s MeSH on demand keyword selector tool to update the list as follows:
“E-waste recyclers, micronutrients, pollution, nutrition, exposure assessment, biomarkers, informal sector, cross-sectional study”.

Point 4: The inclusion criteria of sample population only include two points, which is not sufficient in this study. At the same time, exclusion criteria of sample population should be displayed?. For example, people with organic and chronic diseases, or people who are taking drugs, etc., are included?

Response 4: Thank you for this observation. In our inclusion criteria at Agbogbloshie, we also considered e-waste recyclers who functioned under the management and control of the Chairman for recyclers to allow effective follow-up. Meanwhile, persons with mental or physical disabilities interfering with their ability to understand the informed consent or complete health status measures were ineligible as participants. We therefore include these criteria in our paper in line 107 to 110 as:

“We also considered e-waste recyclers who functioned under the management and control of the Chairman for recyclers to allow effective follow-up. Meanwhile, persons with mental or physical disabilities interfering with their ability to understand the informed consent or complete health status measures were ineligible as participants”.

Point 5: Generally, the 3-day 24-hour retrospective method was used in dietary survey, in which, the survey time should be randomly selected from Monday to Sunday for there may be significant difference in the dietary quantity and quality between working days and rest days and so on.

Response 5: We used a 2-day 24-hour recall retrospective method in the dietary survey as indicated in line 146 of page 4. Specifically dietary data collected was randomly collected on one weekday and one weekend. This was clearly indicated in lines 153 to 156, which read:

“The interview consisted of questions on foods and beverages (e.g., the amount, the time and the types of meals/foods) consumed on one weekday and one day of preceding weekend (Saturday or Sunday). In all cases, information was solicited within less than 24 hours when that day ended”.

Point 6: The correlation coefficient between dietary micronutrient intake and micronutrient levels in blood and urine of e-waste recyclers are relative low, but their statistical difference was significant in table 4?

Response 6: Yes, it is possible to have a low correlation coefficient with a statistically significant p value. In our case, we found a relationship between dietary micronutrient intake and micronutrient levels in blood and urine in some instances, but the relationship was not strong.

Point 7: According to results of this paper, e-waste environmental pollution has little impact on the health of the population?. The most important factors of insufficient micronutrient state of e-waste recyclers and general population is socioeconomic factor? 

Response 7: Many thanks for this point. E-waste contamination can have a major effect on the health of the population to some extent. Apparently, lower levels of Ca, Mg, Zn, Cu, Fe and Se were measured in blood of the e-waste recyclers (Table 3a), suggesting that some level of interaction between micronutrients and toxic metals in the body after exposure may affect their micronutrient status, given that several studies have reported increased exposure among these worker groups to pollutants (Chama et al., 2014; Srigboh et al., 2016; Wittsiepe et al., 2017). We do agree that one key factor for the insufficient micronutrient status of the e-waste recyclers and general population may be the socio-economic factor, however the exposure to toxic metals due to informal recycling of e-waste may also influence the micronutrient status of recyclers and controls.

REFERENCES:

Bonner, F. W. (1981). The acute and subacute effects of cadmium on calcium homeostasis and bone trace metals in the rat. Journal of inorganic biochemistry, 14(2), 107-114.

Brzoska, M. (2000). Effect of short-term ethanol administration on cadmium retention and bioelement metabolism in rats continuously exposed to cadmium. Alcohol and Alcoholism, 35(5), 439-445.

Goyer, R. A. (1997). Toxic and essential metal interactions. Annual review of nutrition, 17(1), 37-50.

Horiguchi, H. (2011). Cadmium induces anemia through interdependent progress of hemolysis, body iron accumulation, and insufficient erythropoietin production in rats. Toxicological Sciences, 122(1), 198-210.

Jan, A. T. (2015). Heavy metals and human health: mechanistic insight into toxicity and counter defense system of antioxidants. International journal of molecular sciences, 16(12), 29592-29630.

Nordberg. (2007). Handbook on the Toxicology of Metals. 3rd Edition, 79-81.

Peraza, M. A. (1998). Effects of micronutrients on metal toxicity. Environmental Health Perspectives, 106(Suppl 1), 203. Retrieved from https://www.ncbi.nlm.nih.gov/pmc/articles/PMC1533267/pdf/envhper00536-0213.pdf

Sears, M. E. (2012). Arsenic, cadmium, lead, and mercury in sweat: a systematic review. Journal of environmental and public health, 2012.

Reviewer 4 Report

This report presents some niche but interesting findings on the micronutrient status of e-waste recyclers in Agbogbloshie, Ghana. 

Specific comments:

  1. How were the participants recruited? I understand they were part of an existing longitudinal study but more details would be helpful.
  2. The choice of the Kruskal-Wallis test was not explained.
  3. No confounder analysis was performed. The role of other confounding factors (e.g. chronic illness) was not appropriately considered, or the role of macronutrients in this context.
  4. To what extent can food-based approaches improve micronutrient status? In addition, potential interactions between micronutrients affecting absorption and bioavailability has to be considered in any supplementation or fortification strategy. This should be at least briefly discussed. 
  5. In terms of areas for future work, 'omics' technology such as non-targeted mass spectrometry-based metabolomics would provide much more information about the effect of chronic micronutrient deficiencies or heavy metal exposure (citation: pubmed.ncbi.nlm.nih.gov/28237781). This should be recommended by the authors.
  6. Authors should try to condense and simplify the tables to improve the general presentation and readability for readers. There are too many tables and numbers at the moment.

Round 2

Reviewer 2 Report

I have no more comment, since the authors have well responsed and revised the manuscript.

Reviewer 3 Report

The authors have revised the manuscript according to the suggestions and there is no other concerns regarding the manuscript.